# Tumor Size >5 cm and Harvested LNs <12 Are the Risk Factors for Recurrence in Stage I Colon and Rectal Cancer after Radical Resection

**DOI:** 10.3390/cancers13215294

**Published:** 2021-10-21

**Authors:** Hye-Sol Jung, Seung-Bum Ryoo, Han-Ki Lim, Min Jung Kim, Sang Hui Moon, Ji Won Park, Seung-Yong Jeong, Kyu Joo Park

**Affiliations:** 1Department of Surgery, Seoul National University Hospital, Seoul National University College of Medicine, Seoul 03080, Korea; 5C741@snuh.org (H.-S.J.); 83212@snuh.org (H.-K.L.); minjungkim@snuh.org (M.J.K.); moon-sanghui@snuh.org (S.H.M.); sowisdom@snuh.org (J.W.P.); syjeong@snu.ac.kr (S.-Y.J.); kjparkmd@plaza.snu.ac.kr (K.J.P.); 2Colorectal Cancer Center, Seoul National University Cancer Hospital, Seoul 03080, Korea; 3Cancer Research Institute, Seoul National University, Seoul 03080, Korea

**Keywords:** stage I colorectal cancer, recurrence, risk factors, survival

## Abstract

**Simple Summary:**

We analyzed the data from 1952 patients with stage I colorectal cancer to evaluate the risk factors for recurrence and survival rates. In the entire cohort, the recurrence rate was 4.6%. There were some differences in the risk factors for recurrence between colon and rectal cancer in stage I colorectal cancer. Left-sided tumors, T2, tumor size >5 cm, and lymphovascular invasion were independent risk factors of colon cancer recurrence. Male, preoperative carcinoembryonic antigen (CEA) ≥2.5 ng/mL, and harvested lymph nodes (LNs) <12 were independently associated with recurrence of rectal cancer. Even though patients with early-stage CRC underwent curative resection, survival sharply decreased in cases of recurrence. Our findings could suggest more aggressive surveillance for patients with an increased risk of recurrence.

**Abstract:**

Recurrence can still occur even after radical resection of stage I colorectal cancer (CRC). This study aimed to identify subgroups with a high risk for recurrence in the stage I CRC. We retrospectively reviewed prospectively collected data of 1952 patients with stage I CRC after radical resection between 2002 and 2017 at our institute. 1398 (colon, 903 (64.6%), rectum, 495 (35.4%)) were eligible for analysis. We analyzed the risk factors for recurrence and survival. During the follow-up period (median: 59 months), 63 (4.6%) had a recurrence. The recurrence rate of rectal cancer was significantly higher than that of colon cancer (8.5% vs. 2.3%). Left-sided tumors, T2, tumor size >5 cm, and lymphovascular invasion were independent risk factors of colon cancer recurrence. Male, preoperative carcinoembryonic antigen (CEA) ≥2.5 ng/mL, and harvested lymph nodes (LNs) <12 were independently associated with recurrence of rectal cancer. Recurrence affected OS (5-year OS: 97.1% vs. 67.6%). Despite curative resection, survival sharply decreased with recurrence. The risk factors for recurrence were different between colon and rectal cancer. Patients with a higher risk for recurrence should be candidates for more aggressive surveillance, even in early-stage CRC.

## 1. Introduction

Colorectal cancer (CRC) is the third most common cancer worldwide [1]. With the recent spread of cancer screening programs and advanced technology, the overall survival (OS) rate of CRC patients has improved in many countries [2]. The reduced mortality rate of CRC may result from the early detection of CRC through screening programs. In Korea, CRC is also increasing and is the second most prevalent cancer. Under the well-organized nationwide CRC screening program in Korea, initiated in 2004, people aged ≥50 years are being screened with yearly fecal occult blood tests. When there is a positive result, a colonoscopy is performed to confirm the disease. The survival rate of CRC patients in Korea has increased in recent years. The rate before 2000 was approximately 50%, which increased to 66.3% between 2001 and 2005, and to approximately 70% between 2004 and 2008 [3]. This improvement might be due to the increasing number of early-stage CRC detected from the national CRC screening programs.

The standard treatment for early-stage CRC is curative resection. Radical colon or rectal resection with regional lymph node (LN) dissection is the treatment of choice, and the 5-year survival rate after curative resection has been reported to be over 90%. However, tumor recurrence can still occur even after radical resection of stage I CRC. Previous studies have reported local recurrence rates from 4.9% to 16.8% in stage I CRC patients [4]. Recurrence of the tumor in early-stage CRC can frustrate patients and surgeons due to failure of long-term recurrence-free survival, in contrast to general expectations. Because adjuvant chemotherapy is usually not recommended for these patients, there is currently no choice but to improve their survival after curative resection. Therefore, it is important to recognize adverse prognostic factors associated with a tumor recurrence, despite the early stage of the tumor and to implement more intense follow-up strategies for patients with high recurrence risk. Some studies have reported clinicopathological factors predicting recurrence in stage I CRC, but comprehensive information has not been provided. 

We aimed to investigate risk factors associated with recurrence of cancer after radical resection and survival outcomes in stage I CRC according to tumor location.

## 2. Materials and Methods

Between 2002 and 2017, 1952 patients with stage I CRC underwent surgical resection at Seoul National University Hospital. We maintained a prospectively collected database with medical records, including data on recurrence and survival during the follow-up period. The clinical characteristics and pathology reports of these patients were retrospectively reviewed. The exclusion criteria were as follows: patients with recurrent CRC, hereditary CRC including familial adenomatous polyposis and hereditary nonpolyposis CRC, local excision or combined synchronous CRC; patients who underwent palliative resection or preoperative concurrent chemoradiation therapy; and patients with incomplete follow-up data. Finally, only 1398 patients were eligible. This study was approved by our institutional review board (H-1910-103-1071). The requirement for written informed consent was waived in this retrospective study. 

Age, sex, preoperative carcinoembryonic antigen (CEA) level, pathologic data including T stage, tumor size, number of harvested LNs, lymphovascular invasion (LVI), perineural invasion (PNI), adjuvant chemo- or radiation therapy, and tumor location were included as variables of interest. The tumor location was divided into the colon and rectum, and the colon was divided into right and left. The right colon included the appendix, cecum, and the ascending and transverse colon; the left colon included the splenic flexure and the descending and sigmoid colon. The tumor was classified as a sigmoid colon tumor when it was located in the rectosigmoid junction. Tumor size was confirmed based on the longest diameter of the tumor recorded in the pathology reports. The pathological tumor stage was determined according to the 7th American Joint Committee on Cancer (AJCC) TNM classification. Local recurrence was defined as tumor recurrence at the anastomosis site or around the region of the primary resection. Distant metastases included distant organ metastases or peritoneal seeding. 

Follow-up after curative resection was performed every 3 or 6 months with the estimation of the serum CEA level, chest radiography, abdominal sonography, and abdominal and pelvic computed tomographic scans. Colonoscopy was performed every 1 or 2 years during the follow-up period. OS was calculated by the time from operation to death, and recurrence-free proportion (RFP) was calculated by the time from operation to the date of tumor recurrence.

Statistical analysis was performed using SPSS version 27.0 for Windows (IBM Corporation, Armonk, NY, USA). Categorical variables were analyzed using the chi-square test and Fisher’s exact test, and continuous variables were compared using the Student’s *t*-test. Multivariate analysis was performed using logistic regression to analyze the risk factors affecting recurrence. RFP and OS were estimated using the Kaplan–Meier method, and the log-rank test was used to compare the differences between the curves. Multivariable analysis of OS and RFP was performed using the Cox regression proportional hazard model. Statistical significance was set at *p* < 0.05. 

## 3. Results

### 3.1. Clinicopathologic Characteristics of the Patients and Risk Factors for Recurrence in Stage I CRC

The mean patient age was 62.7 ± 10.4 years, and 59.9% were male. Colon and rectal cancer occurred in 903 (64.6%) and 495 (35.4%) patients, respectively. All patients underwent R0 resection. Chemotherapy was administered to nine patients (0.6%) because of a close distal resection margin or unfavorable pathological findings, such as LVIs. During the median follow-up period of 59 months (range: 2–84 months), 63 (4.6%) experienced tumor recurrence. Male patients and those with rectal cancer, preoperative CEA levels ≥2.5 ng/mL, a T2 tumor, a tumor size >5 cm, and harvested LNs <12 were more likely to have a recurrence. A mean of 18.9 ± 9.8 LNs (range: 0–97) were retrieved in the enrolled patients. More than 12 LNs were harvested in 1145 patients (77.4%). Details of the clinicopathological characteristics are presented in Table 1.

### 3.2. Risk Factors for Recurrence

Of the 903 patients with colon cancer, 21 (2.3%) experienced recurrence. A univariate analysis showed that having left-sided colon cancers, a T2 tumor, a tumor size >5 cm, and proximal and distal resection margins ≤5 cm were significantly associated with tumor recurrence. Having left-sided colon cancer (odds ratio [OR]: 9.524, 95% confidence interval [CI]: 1.129–80.374, *p* = 0.038), T2 (OR: 3.645, 95% CI: 1.181–11.248, *p* = 0.025), a tumor size >5 cm (OR: 5.124, 95% CI: 1.537–17.082, *p* = 0.008), and LVI (OR: 3.168, 95% CI: 1.076–9.334, *p* = 0.036) were independent risk factors for recurrence in multivariate analysis (Table 2).

Of the 453 patients with rectal cancer, recurrence occurred in 42 (8.5%). Being male and having an elevated CEA level, poorer differentiation, a pT2 tumor, and LNs <12 examined were associated with an increased risk of recurrence in univariate analysis. Among them, being male (OR: 2.564, 95% CI: 1.184–5.551, *p* = 0.017), having an elevated CEA level (OR: 2.010, 95% CI: 1.008–4.008, *p* = 0.047), and having LNs <12 harvested (OR: 2.460, 95% CI: 1.228–4.927, *p* = 0.011) still had a significant impact on recurrence in multivariate analysis (Table 3).

The recurrence rate difference by the operation period (2002–2006, 2007–2011, 2012–2017) was analyzed. In the colon cancer group, using univariate analysis, the recurrence rate was 3.7% in 2002–2006, 2.1% in 2007–2011, and 2.0% in 2012–2017, respectively (*p* = 0.502). In the rectal cancer group, the recurrence rate was significantly different as 13.3%, 8.9%, and 5.0% with univariate analysis, respectively (*p* = 0.045). The statistical significance by the operation period in the rectal cancer group did not present in the multivariate analysis. 

### 3.3. Treatment for Recurrence

Table 4 shows characteristics and the clinical course of patients with recurrence. The median follow-up duration and time to recurrence were 62 months (range: 4–84) and 18 months (range: 3–68), respectively. Locoregional recurrence and distant metastasis occurred in 28 (44.4%) and 28 (44.4%) patients, respectively; both occurred in 7 patients (11.1%). In colon cancer, distant metastasis was more common than locoregional recurrence (locoregional vs. distant, 7 vs. 12), but in rectal cancer, locoregional recurrence was more frequent (locoregional vs. distant, 21 vs. 16). Among the 63 patients with recurrence, 53 (84.1%) underwent surgical resection, and 26 (26/53, 49.1%) received R0 resection. Chemotherapy and radiotherapy were provided to 44 (69.8%) and 12 patients (19.0%), respectively. A total of 19 patients (30.2%) who were followed up had no evidence of disease after treatment for recurrence.

### 3.4. Survival Outcomes

During the follow-up period, 116 patients (8.3%) died, among whom 32 (27.6%) died from recurrence. Patients with rectal cancer (5-year OS: 96.6% vs. 94.0%; *p* = 0.013) (Figure 1A), elevated CEA levels (5-year OS: 96.7% vs. 92.1%, *p* < 0.001), or recurrence (5-year OS: 97.1% vs. 67.6%, *p* < 0.001) (Figure 1B) had a decreased OS rate. Male patients (5-year OS: 97.6% vs. 98.2%, *p* = 0.021) and those with preoperative CEA levels ≥2.5 ng/mL (5-year OS: 97.2% vs. 94.1%, *p* = 0.002) presented significantly lower survival rates in colon cancer. Colon cancer patients with a tumor size >5cm (*p* = 0.089, (Figure 2A) or LVI (*p* = 0.774, (Figure 2B) were likely to have worse outcomes, but the difference was not statistically significant. Elevated preoperative CEA levels (5-year OS: 89.5% vs. 95.6%, *p* = 0.005) (Figure 3A) and fewer harvested LNs (5-year OS: 89.9% vs. 95.3%, *p* < 0.001) (Figure 3B) in the rectal cancer group were significantly associated with worse OS. The univariate and multivariate analysis of RFP and OS in colon and rectal cancer group is described in Table 5 and Table 6.

## 4. Discussion

This single-center cohort study investigated the risk factors for tumor recurrence and the long-term outcomes in stage I CRC. The overall incidence of recurrence in stage I CRC after curative radical resection was 4.6%. Patients with rectal cancer had significantly more recurrence than those with colon cancer, and there were some differences in risk factors for recurrence between colon and rectal cancer. Having left-sided colon cancer, a pT2 tumor, a tumor size >5 cm, or LVI were independent risk factors for recurrence. In rectal cancer, male sex, preoperative CEA levels ≥2.5 ng/mL, or harvested LNs <12 were independent risk factors for recurrence. Recurrence had a significant impact on survival outcomes.

Even in early-stage CRC without LN metastases, there can still be recurrence in some patients. Although metastatic LNs were not detected on pathologic examination, there is a possibility of micrometastases within regional LNs [5], skip metastases to distant LNs [6], or even hematogenous metastases [7]. The micrometastases are too small to be detected using routine histologic examination. A previous study using immunohistochemistry to detect such lesions reported that micrometastasis was identified in 67% of the recurrence group and 84% of the disease-free group in pT3 or pT4 colon cancers. Some authors reported that this was found in approximately 3% of patients (2.5–3.5%), even in stage I-II rectal cancer [8]. Skip metastases to distant LNs have been reported in 20–34% of advanced CRC in molecular analysis [9], and hematogenous metastases may also result in the recurrence of the tumor, despite node-negativity [10]. Circulating tumor DNA, which is a fragmented cell-free DNA derived from tumor cells and a novel prognostic marker for recurrence, was detected in stage I CRC, although the concentration was lower than that in advanced diseases [11]. These findings imply that the recurrence of node-negative CRC may result from hematogenous metastasis of circulating tumor cells already existing at diagnosis, even in the early-stage. Herein, we noted locoregional recurrence as well as distant metastasis, which presented with poor survival outcomes, and it is important to identify the risk factors related to recurrence, even in stage I CRC. In high-risk stage II CRC, adjuvant therapy can provide some benefits; however, there has been no evidence of additional treatments in stage I CRC [12]. We primarily aimed to determine high-risk subgroups with analysis using the patients’ clinical and pathological information, thus that oncologic outcome can be improved for these high-risk patients with a more intensive follow-up or additional adjuvant therapies. Although the survival outcomes after recurrence were significantly lower even in early stage, some of risk factors associated with recurrence were not exactly matched with survival outcomes in this study. It might be possible that the recurrence rate was not high, and long-term follow-up might attenuate the effects on survival outcomes.

Having a tumor size >5 cm was an independent risk factor for recurrence in stage I colon cancer in this study. Patients with larger tumors seemed to have a lower OS, but this was not statistically significant. Large tumors may have a more unfavorable underlying tumor biology. Some studies described the correlation of larger tumor size with an advanced T stage, more nodal invasion, poorer differentiation, tumor necrosis, and the presentation of higher vascular endothelial growth factor levels [13]. A large cohort study revealed that more than 50% of patients with tumors >4 cm were node-positive, and tumor size was significantly associated with OS [14]. Colon cancer with a large tumor size could be underestimated by evaluating the T stage, as more efforts may be needed to find the tumor cell invasions beyond proper muscle on pathologic examination. Although there was no LN positivity, we can also assume that there is a higher chance of hematogenous spread in larger tumors. Patients with large tumors may benefit from adjuvant therapy after surgical resection, and further studies are necessary. 

LVI was related to the recurrence of colon cancer in our multivariate analysis. It has been widely recognized as a poor prognostic factor related to lymphatic metastasis in early-stage CRC [15]. A recent study reported that 8.5% of patients with stage I colon cancer had LVI [16]. Some studies showed that tumors with LVI have a higher likelihood of advanced T stage or tumor budding and infiltrating or poorly differentiated histology [17]. LVI has also been suggested as a risk factor for micrometastases or skip metastasis [18], and these may explain the metastatic potential of LVI, even in patients without LN metastases. Distant metastasis tended to be more dominant than local recurrence in colon cancer.

For early-stage colon cancers, there is a lack of evidence regarding recurrence by tumor laterality. Traditionally, right-sided colon cancer has been reported to have poor oncologic outcomes because of detection at more advanced stages, larger tumor size, aggressive tumor biology with poorly differentiated histology, and particular tumor development from epigenetic aberrations with DNA hypermethylation of the mismatch repair gene [19]. However, in the analysis according to the stages, there was no difference in oncologic outcomes between right- and left-sided colon cancer. A recent study revealed that OS was similar between left- and right-sided colon cancer patients with stage I and II disease, and the recurrence rate did not differ between right- and left-sided colon cancers in stage I [20]. Left-sided colon cancers recurred more frequently in this study; however, this did not affect the survival compared to right-sided colon cancers. It has been considered that rectosigmoid colon cancer is likely to behave as rectal cancer [21], and postoperative complications such as anastomosis leakage or pelvic abscess could lead to poor oncologic outcomes [22]. Furthermore, operative procedures might be more complex in descending colon cancer. Although complete mesocolic excision has been a standard procedure in colon cancer surgery, it has not yet been established in descending colon cancer [23,24], and there has been some debate regarding the optimal extent of LN dissection, including ligation of the inferior mesenteric artery or vein [25]. It was also considered that recurrence developed in a small number of patients in this study, and it might not be enough to evaluate the long-term survival. 

In this study, tumor location in the rectum was a worse prognostic factor for recurrence and OS in stage I CRC, consistent with previous studies. Owing to the limitation of the operating field, total mesorectal excision (TME) has been considered a challenging procedure, and TME quality is the most important factor to prevent local recurrence [26]. The surgical plane is more difficult to maintain in men due to a narrow pelvis, which is consistent with a previous study that reported that worse outcomes in men might be associated with difficulty in obtaining adequate lateral resection margins [27]. Our results showed that locoregional recurrence was more frequent in rectal cancer and male patients experienced more recurrences. Different lymphatic or venous drainage systems along the mesorectum and internal iliac vessels could make the surgery more complex. A study revealed that some patients with early-stage rectal cancer with a high risk of recurrence would benefit from preoperative radiotherapy [28]. 

It is well known that rectal cancer patients with a smaller number of harvested LNs have a higher recurrence rate and worse survival outcomes. Because of lower LN chains in the mesorectum with the difficulty of TME, there might be a higher risk of insufficient retrieval of the LNs in patients with rectal cancer [29]. In these cases, there might also be a possibility of underestimation of pathological staging [30], which could present as tumor recurrence even in early-stage rectal cancer. Furthermore, if LNs were not sufficient for evaluation, we cannot exclude the possibility of micrometastases or skip lesions in the perirectal or pelvic LNs that were not detected with our pathologic examinations. Since adequate retrieval of LNs is associated with surgical radicality, an individual surgeon’s skill and insufficient node dissection would mainly impact the oncologic outcomes, especially in rectal cancer. Our results showed that approximately 20% of rectal cancer patients had <12 harvested LNs. This criterion was developed mostly in early 2000, when the clinical guidelines began to recommend obtaining sufficient LNs, and our findings did not seem to be worse than those reported in previous studies [31]. This study presented the recurrence rate of stage I rectal cancer significantly decreased over the past 15 years of this study period. When comparing the lymph node yield, the incidence of lymph node yield <12 significantly decreased by the period as 56.6% in 2002–2007, 12.4% in 2008–2012, and 4.4% in 2013–2017 (*p* < 0.001). The poor lymph node yield, or an inadequate lymph node examination due to lack of recognition for the importance of radicality based on lymph node retrieval may cause an underestimation of stage in early-stage CRC although the total mesorectal excision or high ligation of inferior mesenteric artery was performed. However, in colon cancer, the recurrence rate did not differ by the operation period. This may result from the no difference in lymph node retrieval and standardization of surgical techniques in colon cancer surgery. 

We also verified that a preoperative CEA level ≥2.5 ng/mL was an independent risk factor of recurrence and worse survival in stage I CRC. However, the reference range of CEA was set at <5 ng/mL in previous studies [32], we determined the cut-off value of CEA as 2.5 ng/mL to facilitate statistical analysis considering early-stage cancer. A recent study suggested that patients with high preoperative CEA levels in stage I-III rectal cancer had a high risk of early locoregional relapse and recurrence of distant metastasis after resection. Our results suggest that an elevated preoperative CEA level is an important prognostic factor, even in stage I cancer. Postoperative elevated CEA levels are a widely known predictive marker for recurrence in CRC. Thus, both preoperative and postoperative measurements of serum CEA levels may provide useful information to predict prognosis.

Our study has several limitations. First, this is a retrospective single-center study; the selection biases could not be avoided. Second, in our study population, some patients underwent adjuvant chemotherapy or radiation therapy despite having stage I cancer. These procedures were performed at the surgeon’s request based on close resection margins. This might have resulted in confusion regarding the influence on oncologic outcomes. Third, we have a prospective cohort from a large number of patients with long-term follow-up, but the eligible number of patients in this study decreased after excluding some patients to reduce the confounding factors. Nevertheless, our study’s strengths are that it can help minimize surgeon-related factors, which is one of the advantages of single-center studies, and it had a relatively large sample size with long-term oncologic outcomes, as well as the identification of different risk factors for recurrence in stage I CRC. The in-depth investigation for the tumor biologies of these patients who experience a tumor recurrence despite the early stage of cancer would help comprehend the disease and lead to better clinical management.

## 5. Conclusions

Being male and having rectal cancer, an elevated preoperative CEA level, a pT2 tumor, a larger tumor size, and a smaller number of LNs harvested were associated with recurrence in stage I CRC. There were some differences in the risk factors for recurrence between the colon and rectal cancers in stage I CRC. Although patients with early-stage CRC underwent curative resection, survival sharply decreased in cases of recurrence. Our findings could provide insight on choosing aggressive surveillance and the necessity of adjuvant treatment for patients with an increased risk of recurrence.

## Figures and Tables

**Figure 1 cancers-13-05294-f001:**
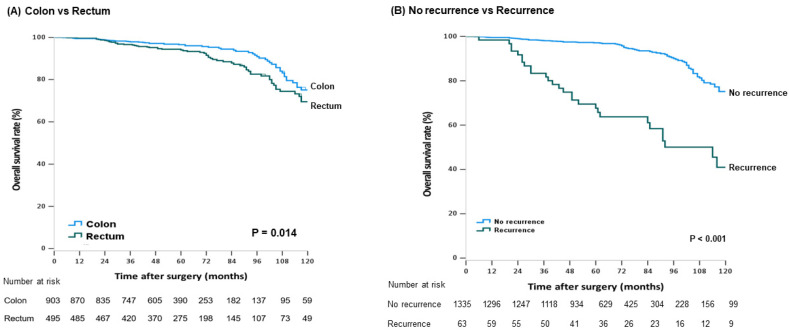
Kaplan–Meier curves for survival analysis with stage I colorectal cancer patients. The overall survival according to tumor location (**A**) and recurrence (**B**). (**A**) Colon vs. rectum, (**B**) no recurrence vs. recurrence.

**Figure 2 cancers-13-05294-f002:**
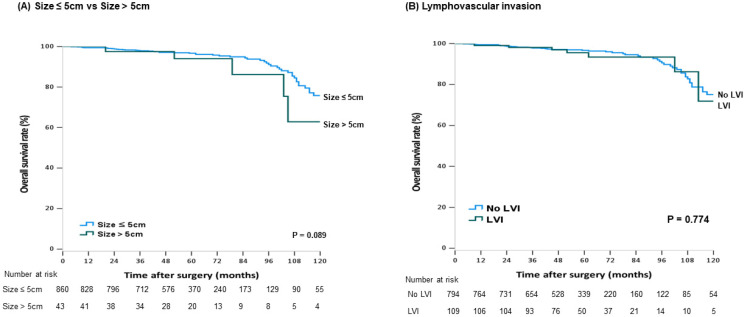
Kaplan–Meier curves for survival analysis with stage I colon cancer patients. The overall survival according to tumor size (**A**) and the presence of lymphovascular invasion (LVI) (**B**). (**A**) Size ≤ 5 cm vs. Size > 5 cm, (**B**) Lymphovascular invasion.

**Figure 3 cancers-13-05294-f003:**
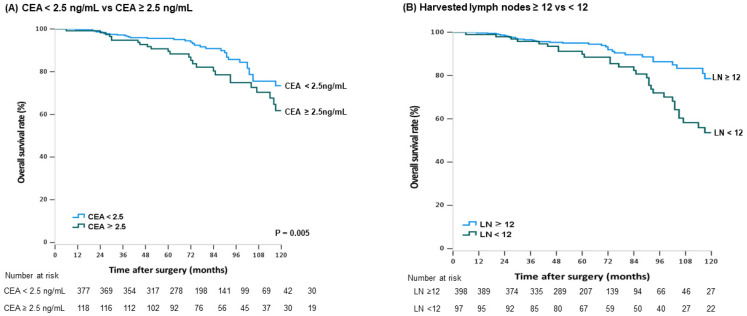
Kaplan–Meier curves for survival analysis with stage I rectal cancer patients. The overall Scheme 2.5 ng/mL vs. (**A**) CEA level ≥2.5 ng/mL, (**B**) harvested lymph nodes ≥12 vs. <12.

**Table 1 cancers-13-05294-t001:** Basic clinicopathological characteristics of colorectal cancer patients.

Variation	Total (*N* = 1398)	No Recurrence (*N* = 1335)	Recurrence (*N* = 63)	*p*-Value
Age (years)	62.7 ± 10.5	62.7 ± 10.45	62.7 ± 8.9	0.970
Sex				* 0.003
Female	561 (40.1)	547 (97.5)	14 (2.5)	
Male	837 (59.9)	788 (94.1)	49 (5.9)	
Location				* <0.001
Colon	903 (64.6)	882 (97.7)	21 (2.3)	
Rectum	495 (35.4)	453 (91.5)	42 (8.5)	
Preoperative CEA (ng/mL)				* 0.037
<2.5	1082 (77.4)	1040 (96.1)	42 (3.9)	
≥2.5	316 (22.3)	295 (93.4)	21 (6.6)	
Differentiation				0.201
WD	327 (23.4)	316 (96.6)	11 (3.4)	
MD	1054 (5.4)	1004 (95.3)	50 (4.7)	
PD	17 (1.2)	15 (88.2)	2 (11.8)	
T stage				* <0.001
T1	625 (44.7)	612 (97.9)	13 (2.1)	
T2	773 (55.3)	723 (93.5)	50 (6.5)	
Size (cm)				* 0.013
≤5	1326 (94.8)	1271 (95.9)	55 (4.1)	
>5	72 (5.2)	64 (88.9)	8 (11.1)	
Harvested LNs				* 0.001
<12	253 (18.1)	232 (91.7)	21 (8.3)	
≥12	1145 (81.9)	1103 (96.3)	42 (3.7)	
LVI				0.052
No	1203 (86.1)	1154 (95.9)	49 (4.1)	
Yes	195 (13.9)	181 (92.8)	14 (7.2)	
PNI				0.276
No	1349 (96.5)	1290 (95.6)	59 (4.4)	
Yes	49 (3.5)	45 (91.8)	4 (8.2)	
Adjuvant chemotherapy				1.000
No	1390 (99.4)	1326 (95.5)	63 (4.5)	
Yes	9 (0.6)	9 (100)	0 (0)	

CEA, carcinoembryonic antigen; WD, well-differentiated; MD, moderately differentiated; PD, poorly differentiated; LN, lymph node; LVI, lymphovascular invasion; PNI, perineural invasion. * *p* < 0.05.

**Table 2 cancers-13-05294-t002:** Univariate and multivariable analysis of colon cancer patients.

Colon (*N* = 903)	Univariate Analysis			Multivariate Analysis	
	No Recurrence (N = 882)	Recurrence (N = 21)	*p*-Value	Odds Ratio (95% CI)	*p*-Value
Age (years)	63.3 ± 9.92	65.9 ± 8.40	0.245		
Sex			0.112		
Female	362 (98.6)	5 (1.4)			
Male	520 (97.0)	16 (3.0)			
Preoperative CEA (ng/mL)			0.747		
<2.5	688 (97.6)	17 (2.4)			
≥2.5	194 (98.0)	4 (2.0)			
Location			* 0.002		
Rt	330 (99.7)	1 (0.3)		1	
Lt	552 (96.5)	20 (3.5)		9.524 (1.129–80.374)	* 0.038
Differentiation			0.839		
WD	228 (97.9)	5 (2.1)			
MD	642 (97.6)	16 (2.4)			
PD	12 (100)	0 (0)			
T stage			* 0.007		
T1	432 (99.1)	4 (0.9)		1	
T2	450 (96.4)	17 (3.6)		3.645 (1.181–11.248)	* 0.025
Size (cm)			* 0.015		
≤5	843 (98.0)	17 (2.0)		1	
>5	39 (90.7)	4 (11.4)		5.124 (1.537–17.082)	* 0.008
PRM (cm)			* 0.034		
>5	747 (98.2)	14 (1.8)		1	
≤5	135 (95.1)	7 (4.9)		1.924 (0.725–5.103)	0.189
DRM (cm)			* 0.046		
>5	487 (98.6)	7 (1.4)		1	
≤5	395 (96.6)	14 (3.4)		1.247 (0.462–3.362)	0.663
Harvest LNs			0.387		
LN < 12	151 (96.8)	5 (3.2)			
LN ≥ 12	731 (97.9)	16 (2.1)			
LVI			0.163		
No	778 (98.0)	16 (2.0)		1	
Yes	104 (95.4)	5 (4.6)		3.168 (1.076–9.334)	* 0.036
PNI			0.462		
No	857 (97.7)	20 (2.3)			
Yes	25 (96.2)	1 (3.8)			
Adjuvant chemotherapy			1.000		
No	878 (97.7)	21 (2.3)			
Yes	4 (100)	0 (0)			

CEA, carcinoembryonic antigen; WD, well-differentiated; MD, moderately differentiated; PD, poorly differentiated; LN, lymph node; LVI, lymphovascular invasion; PNI, perineural invasion; PRM, proximal resection margin; DRM, distal resection margin; CI, confidence interval. * *p* < 0.05.

**Table 3 cancers-13-05294-t003:** Univariate and multivariable analysis of rectal cancer patients.

Rectum (*N* = 495)	Univariate Analysis			Multivariate Analysis	
	No Recurrence (*N* = 453)	Recurrence (*N* = 42)	*p*-Value	Odds Ratio (95% CI)	*p*-Value
Age (years)	61.4 ± 11.4	61.1 ± 8.9	0.853	-	-
Sex			* 0.014		
Female	185 (95.4)	9 (4.6)		1	
Male	268 (89.0)	33 (11.0)		2.564 (1.184–5.551)	* 0.017
Preoperative CEA (ng/mL)			* 0.008		
<2.5	352 (93.4)	25 (6.6)		1	
≥2.5	101 (85.6)	17 (14.4)		2.010 (1.008–4.008)	* 0.047
Differentiation			* 0.031		
WD	88 (93.6)	6 (6.4)		1	
MD	362 (91.4)	34 (8.6)		1.269 (0.498–3.237)	0.618
PD	3 (60.0)	2 (40.0)		6.385 (0.780–52.255)	0.084
T stage			* 0.019		
T1	180 (95.2)	9 (4.8)		1	
T2	273 (89.2)	33 (10.8)		1.938 (0.872–4.306)	0.104
Size (cm)			0.842		
≤5	414 (91.6)	38 (8.4)			
>5	39 (90.7)	4 (9.3)			
Harvest LNs			* 0.002		
LN ≥ 12	372 (93.5)	26 (6.5)		1	
LN < 12	81 (83.5)	16 (16.5)		2.460 (1.228–4.927)	* 0.011
PRM (cm)			0.153		
>5	427 (91.0)	42 (9.0)			
≤5	26 (100)	0 (0)			
DRM (cm)			0.243		
>1	226 (93.0)	17 (7.0)			
≤1	227 (90.1)	25 (9.9)			
CRM (cm)			1.000		
>1	360 (90.9)	36 (9.1)			
≤1	6 (100)	0 (0)			
NA	87 (93.5)	6 (6.5)			
LVI			0.468		
No	376 (91.9)	33 (8.1)			
Yes	77 (89.5)	9 (10.5)			
PNI			0.432		
No	433 (91.7)	39 (8.3)			
Yes	20 (87.0)	3 (13.0)			
Adjuvant chemotherapy			1.000		
No	448 (91.4)	42 (8.6)			
Yes	5 (100)	0 (0)			
Adjuvant radiation therapy			0.057		
No	425 (92.2)	36 (7.8)			
Yes	28 (82.4)	6 (17.6)			

CEA, carcinoembryonic antigen; WD, well-differentiated; MD, moderately differentiated; PD, poorly differentiated; LN, lymph node; LVI, lymphovascular invasion; PNI, perineural invasion; PRM, proximal resection margin; DRM, distal resection margin; CRM, circumferential resection margin; CI, confidence interval. * *p* < 0.05.

**Table 4 cancers-13-05294-t004:** Details of patients with recurrence (*n* = 63).

Variables	*N* (%)
Follow-up duration (median, month)	62 (4–84)
Time to recurrence (median, month)	18 (3–68)
Patterns of recurrence	
Locoregional	28 (44.4)
Distant	28 (44.4)
Both	74 (11.1)
Locoregional recurrence	
Abdominal cavity	24 (38.1)
Anastomosis site	13 (20.6)
Distant metastasis	
Liver	23 (36.5)
Lung	10 (15.9)
Treatment for recurrence	
Surgery and Chemotherapy ± Radiotherapy	36 (57.1)
Surgery only	7 (11.1)
Chemotherapy only	9 (14.3)
Supportive care	10 (15.9)
Follow-up loss	1 (1.6)
Surgery for recurrence	
R0 resection	50 (77.4)
R1 resection	7 (13.2)
R2 resection	5 (9.4)
No evidence of disease after treatment for recurrence	
Yes	19 (30.2)
No	36 (57.1)
Follow-up loss	8 (12.7)

**Table 5 cancers-13-05294-t005:** Univariate and multivariate analysis of risk factors for recurrence-free proportion and overall survival in colon cancer group.

Colon Cancer	Recurrence-Free Proportion				Overall Survival			
	5-YR RFP (%)	*p*-Value	HR (95% CI)	*p*-Value	5-YR OS (%)	*p*-Value	HR (95% CI)	*p*-Value
Sex		0.122	-	-		0.021		0.035
Male	96.2				95.4		1	
Female	98.5				98.2		0.536 (0.301–0.957)	
Preoperative CEA (ng/mL)		0.766	-	-		0.002		0.005

<2.5	97.1				97.2		1	
≥2.5	97.0				94.1		2.028 (1.237–3.322)	
Location		0.003		0.019		0.893	-	-
Rt	99.7		1		97.3			
Lt	95.8		11.044 (1.481–82.342)		96.2			
Differentiation		-	-	-		-	-	-
WD	97.5				95.1			
MD	96.9				97.0			
PD	-				-			
T stage		0.009		0.033		0.319	-	-
T1	98.8		1		95.1			
T2	95.6		3.299 (1.100–9.892)		97.8			
Size (cm)		0.001		0.011		0.089	-	-
≤5	97.6		1		96.7			
>5	88.2		4.175 (1.391–12.524)		94.0			
Harvest LN		0.545	-	-		0.482	-	-
LN ≥ 12	97.5				97.0			
LN < 12	96.0				94.7			
LVI		0.119	-	-		0.774	-	-
No	97.8				96.7			
Yes	93.0				95.6			
PNI		0.601	-	-		-	-	-
No	97.2				96.4			
Yes	94.4				-			

CEA, carcinoembryonic antigen; WD, well-differentiated; MD, moderately differentiated; PD, poorly differentiated; LN, lymph node; LVI, lymphovascular invasion; PNI, perineural invasion; CI, confidence interval.

**Table 6 cancers-13-05294-t006:** Univariate and multivariate analysis of risk factors for recurrence-free proportion and overall survival in rectal cancer group.

Rectal Cancer	Recurrence-Free Proportion				Overall Survival			
	5-YR RFP (%)	*p*-Value	HR (95% CI)	*p*-Value	5-YR OS (%)	*p*-Value	HR (95% CI)	*p*-Value
Sex		0.019		0.024		0.036		0.031
Male	87.8		1		91.3		1	
Female	94.3		0.428 (0.204–0.895)		98.1		0.551 (0.320–0.948)	
Preoperative CEA (ng/mL)		0.014		0.058		0.008		0.030
<2.5	92.1		1		95.3		1	
≥2.5	84.9		1.842 (0.981-3.459)		89.5		1.680 (1.052–2.682)	
Differentiation		0.013	1	0.291		0.604	-	-
WD	89.6				97.7			
MD	90.7				93.2			
PD	60.0				80.2			
T stage		0.022		0.103		0.175	-	-
T1	94.2		1		95.6			
T2	88.0		1.885 (0.879–4.042)		92.8			
Size (cm)		0.957				0.950	-	-
≤5	90.4				93.9			
>5	89.4				92.9			
Harvest LN		0.005		0.022		0.001		0.003
LN ≥ 12	92.7		1		95.0		1	
LN < 12	82.3		2.090 (1.113–3.923)		89.9		2.063 (1.281–3.321)	
PRM (cm)		-	-	-		-	-	-
>5	89.7				93.5			
≤5	-				-			
DRM (cm)		0.251	-	-		0.111	-	-
>1	91.8				94.1			
≤1	88.9				93.6			
CRM (cm)		-	-	-		-	-	-
>1	89.9				93.8			
≤1	-				-			
LVI		0.416	-	-		0.961	-	-
No	90.7				94.4			
Yes	88.1				91.0			
PNI		0.452	-	-		0.886	-	-
No	90.6				94.0			
Yes	86.7				90.9			

CEA, carcinoembryonic antigen; WD, well-differentiated; MD, moderately differentiated; PD, poorly differentiated; LN, lymph node; LVI, lymphovascular invasion; PNI, perineural invasion; PRM, proximal resection margin; DRM, distal resection margin; CRM, circumferential resection margin; CI, confidence interval.

## Data Availability

The data presented in this study are available on request from the corresponding author. The data are not publicly available due to the permission issue of participants.

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
