# Peer review of "Tumor Size >5 cm and Harvested LNs <12 Are the Risk Factors for Recurrence in Stage I Colon and Rectal Cancer after Radical Resection"

_cancers, 2021, doi:10.3390/cancers13215294_

Round 1
Reviewer 1 Report
I found that it was a well written and interesting paper.
I have noticed one issue you need to adress: You write in the methods section that you have used Cox proportional Hazard model to investigate both survival and recurrence. I cannot find the results of these analyses. I think it would be a good idea to present those results and then maybe leave out some of the Kaplan Meier plots instead.
A small comment: I do not understand what you mean in the introduction section lines 60-61.
I am looking forward to reading the revised manuscript.
Author Response
Reviewer 1
I found that it was a well written and interesting paper.
I have noticed one issue you need to adress: You write in the methods section that you have used Cox proportional Hazard model to investigate both survival and recurrence. I cannot find the results of these analyses. I think it would be a good idea to present those results and then maybe leave out some of the Kaplan Meier plots instead.
Response > Thank you for your kind comments making our manuscript more comprehensive. We used the Kaplan-Meier method to estimate the RFS and OS for univariate analysis and CoX-regression proportional Hazard model for multivariable analysis. We had described the overall survival curve for some variables with significant clinical meaning by the Kaplan Meier curves, but added all of the univariate and multivariate analysis of risk factors for the recurrence-free and overall survival in Table 5 and 6, as you recommended.
Table 5. Univariate and multivariate analysis of risk factors for recurrence-free survival and overall survival in colon cancer group.
Colon cancer |
Recurrence-free survival |
|
Overall survival |
|
|||||
|
5-YR RFS (%) |
p-value |
HR (95% C.I.) |
p-value |
5-YR OS(%) |
p-value |
HR (95% C.I.) |
p- value |
|
Sex Male Female |
96.2 98.5 |
0.122 |
- |
- |
95.4 98.2 |
0.021 |
1 0.536(0.301-0.957) |
0.035 |
|
Preoperative CEA (ng/ml) < 2.5 ≥ 2.5 |
97.1 97.0 |
0.766 |
- |
- |
97.2 94.1 |
0.002 |
1 2.028(1.237-3.322) |
0.005 |
|
Location Rt Lt |
99.7 95.8 |
0.003 |
1 11.044(1.481-82.342) |
0.019 |
97.3 96.2 |
0.893 |
- |
- |
|
Differentiation WD MD PD |
97.5 96.9 - |
- |
- |
- |
95.1 97.0 - |
- |
- |
- |
|
T stage T1 T2 |
98.8 95.6 |
0.009 |
1 3.299(1.100-9.892) |
0.033 |
95.1 97.8 |
0.319 |
- |
- |
|
Size (cm) ≤ 5 > 5 |
97.6 88.2 |
0.001 |
1 4.175(1.391-12.524) |
0.011 |
96.7 94.0 |
0.089 |
- |
- |
|
Harvest LN LN ≥ 12 LN < 12 |
97.5 96.0 |
0.545 |
- |
- |
97.0 94.7 |
0.482 |
- |
- |
|
LVI No Yes |
97.8 93.0 |
0.119 |
- |
- |
96.7 95.6 |
0.774 |
- |
- |
|
PNI No Yes |
97.2 94.4 |
0.601 |
- |
- |
96.4 - |
- |
- |
- |
CEA, carcinoembryonic antigen; WD, well-differentiated; MD, moderately differentiated; PD, poorly differentiated; LN, lymph node; LVI, lymphovascular invasion; PNI, perineural invasion; CI, confidence interval
Table 6. Univariate and multivariate analysis of risk factors for recurrence-free survival and overall survival in rectal cancer group.
Rectal cancer |
Recurrence-free survival |
|
Overall survival |
|
|
||||
|
5-YR RFS (%) |
p-value |
HR (95% C.I.) |
P-value |
5-YR OS (%) |
p-value |
HR (95% C.I.) |
P -value |
|
Sex Male Female |
87.8 94.3 |
0.019 |
1 0.428(0.204-0.895) |
0.024 |
91.3 98.1 |
0.036 |
1 0.551(0.320-0.948) |
0.031 |
|
Preoperative CEA (ng/ml) < 2.5 ≥ 2.5 |
92.1 84.9 |
0.014 |
1 1.842(0.981-3.459) |
0.058 |
95.3 89.5 |
0.008 |
1 1.680(1.052-2.682) |
0.030 |
|
Differentiation WD MD PD |
89.6 90.7 60.0 |
0.013 |
1
|
0.291 |
97.7 93.2 80.2 |
0.604 |
- |
- |
|
T stage T1 T2 |
94.2 88.0 |
0.022 |
1 1.885(0.879-4.042) |
0.103 |
95.6 92.8 |
0.175 |
- |
- |
|
Size (cm) ≤ 5 > 5 |
90.4 89.4 |
0.957 |
|
|
93.9 92.9 |
0.950 |
- |
- |
|
Harvest LN LN ≥ 12 LN < 12 |
92.7 82.3 |
0.005 |
1 2.090(1.113-3.923) |
0.022 |
95.0 89.9 |
0.001 |
1 2.063(1.281-3.321) |
0.003 |
|
PRM (cm) >5 ≤ 5 |
89.7 - |
- |
- |
- |
93.5 - |
- |
- |
- |
|
DRM (cm) >1 ≤ 1 |
91.8 88.9 |
0.251 |
- |
- |
94.1 93.6 |
0.111 |
- |
- |
|
CRM (cm) >1 ≤ 1 |
89.9 - |
- |
- |
- |
93.8 - |
- |
- |
- |
|
LVI No Yes |
90.7 88.1 |
0.416 |
- |
- |
94.4 91.0 |
0.961 |
- |
- |
|
PNI No Yes |
90.6 86.7 |
0.452 |
- |
- |
94.0 90.9 |
0.886 |
- |
- |
|
CEA, carcinoembryonic antigen; WD, well-differentiated; MD, moderately differentiated; PD, poorly differentiated; LN, lymph node; LVI, lymphovascular invasion; PNI, perineural invasion; PRM, proximal resection margin; DRM, distal resection margin; CRM, circumferential resection margin; CI, confidence interval
A small comment: I do not understand what you mean in the introduction section lines 60-61.
Response > We agree that the sentence in line 60-61 is confusing. The sentence was redescribed as follows : “it is important to recognize adverse prognostic factors associated with a tumor recurrence, despite the early stage of the tumor”

Reviewer 2 Report
Dear Authors,
The manuscript is interesting and clinically relevant. However, some aspects should be improved:
- The title is too long, and do not bring any novelty.
- An important issue is the definition of Stage I CRC. In stage I, which was the stage used in this study. if the cancer has grown through the muscularis mucosa into the submucosa- T1; and it may also have grown into the muscularis propria-T2, which is in accordance with the manuscript. Regardind the lymph node, in stage I CRC do not spread to nearby lymph nodes (N0) or to distant sites (M0). However, in table 2, the authors approach LN in two categories: "Harvest LN" and "LVI". Please comment on this.
- A major limitation is indeed clinical outcome. It is very confusing. The results can be better explored.
- The cohort is very robust (1952 patients) but when stratified with ou without recurrence, the numbers decrease and there´s no statistical significance.
Author Response
Reviewer 2
The manuscript is interesting and clinically relevant. However, some aspects should be improved:
- The title is too long, and do not bring any novelty.
- Response > We appreciate your precise and considerate reviews. The new finding in this study was not so many but tumor size was the significant risk factors for recurrence in colon There was a few studies to verify the risk of tumor size in stage I colon cancer. Colon cancer with a large tumor size could be underestimated by evaluating the T stage, as more efforts may be needed to find the tumor cell invasions on pathologic examination. Although there was no LN positivity, we can also assume that there is a higher chance of hematogenous spread in larger tumors. We think those were our novelty and we retitled this study “Tumor size or harvested LNs<12 are the risk factors for recurrence in stage I colon and rectal cancer after radical resection”.
- An important issue is the definition of Stage I CRC. In stage I, which was the stage used in this study. if the cancer has grown through the muscularis mucosa into the submucosa- T1; and it may also have grown into the muscularis propria-T2, which is in accordance with the manuscript. Regardind the lymph node, in stage I CRC do not spread to nearby lymph nodes (N0) or to distant sites (M0). However, in table 2, the authors approach LN in two categories: "Harvest LN" and "LVI". Please comment on this.
- Response > We agree that the definition of stage I CRC according to the AJCC TNM classification present no lymph nodes metastasis as you commented. However, we aimed to investigate the comprehensive pathologic data of stage I CRC following the radical resection. The radical resection includes the procedures ligating the origin of feeding vessels and taking the all pericolic and regional lymph nodes. Thus, the lymph node yield is an important surrogate for the radicality of surgery. Furthermore, the possibility of skip metastasis or hematogenous spread of tumor cells may explain the recurrence in the node-negative colorectal cancer stage. In this context, we approached LN in “harvest LN” and “LVI” as an factors harboring recurrence risks in this study. The basic pathologic reports following the radical resection provided by our pathologists also include the number of examined (harvested) lymph nodes and presence of LVI even in the stage I CRC.
- A major limitation is indeed clinical outcome. It is very confusing. The results can be better explored.
- Response > Treatment of stage I colon and rectal cancer is similar and radical resection is standard treatment. We thought the risk of recurrence would be similar between colon and rectal cancer, but we resulted in the difference between colon and rectum. The important novel findings in this study are the tumor size in colon cancer and harvested LNs<12 were the significant risk factors in Stage I. These mean the underestimated tumor stage might be possible and we have to follow-up these patients more aggressively. We investigated the risk factors for a tumor recurrence in the entire cohort (Table 1), and then in the stage I colon (Table 2) and rectal (Table 3) cancer patients cohort. There were some differences in the risk factors for recurrence between colon (tumor sidedness, T stage, tumor size, LVI) and rectal (male, high preoperative CEA level, harvest LN <12) cancer in stage I colorectal cancer. And then, the subgroup analysis for the patients with recurrence (n=63) was performed and presented in table 4. The Kaplan-Meier curves considered most confusing was deleted and we added the comprehensive survival analysis data in Table 5 & 6. Survival outcomes were not matched with risk factors of recurrence. I think it’s because the recurrence rate was not high and long-term follow-up more than 10 years presented tendency to natural course of aging effects. We added those confusing results between risk factors of recurrence and survival outcomes in discussion as your advice.
Line 241-244
“Although the survival outcomes after recurrence were significantly lower even in early stage, some of risk factors associated with recurrence were not exactly matched with survival outcomes in this study. It might be possible that the recurrence rate were not high and long-term follow-up might attenuate the effects on survival outcomes.”
- The cohort is very robust (1952 patients) but when stratified with ou without recurrence, the numbers decrease and there´s no statistical significance.
Response > I totally agree with you and about 2000 patients of a single center cohort during 15 years is large and robust. We maintained a prospectively collected database for about 20 years from our electronic medical records. But, unfortunately, to reduce confounding factors, we have to exclude the patients with recurrent CRC, hereditary CRC including familial adenomatous polyposis and hereditary nonpolyposis CRC, local excision or combined synchronous CRC, patients who underwent palliative resection or preoperative concurrent chemoradiation therapy and patients with incomplete follow-up data. The eligible number of patients was 1,398, and these are also large number of cohort in a single center. There are some advantages for single center large cohort with long-term follow-up compared to the studies from the multicenter big data. The surgical procedure is standardized and almost similar according to the surgeons and pathologic report is also standardized. The clinical variables almost exist and empty data is very rare. However, we described the limitation of our reduced number of the patients in our cohort as you mentioned in Line 344-346:
“Third, we have a prospective cohort from large number of patients with long-trem follow-up, but the eligible number of patients in this study decreased after excluding some patients to reduce the confounding factors.” Nevertheless, our study’s strengths are that it can help minimize surgeon-related factors, which is one of the advantages of single-center studies, and it had a relatively large sample size with long-term oncologic outcomes, as well as the identification of different risk factors for recurrence in stage I CRC.

Reviewer 3 Report
Authors assess the relationship between clinicopathological factors and colorectal cancer recurrence instage 1 disease. This article is sound in its design and limited by the single centre nature of the work, as appreciated by the authors themselves. The findings of larger tumours, poor lymph node yield (a surrogate for inadequate surgery), and differentiation are not surprising.
Comments: Are there any predictors of local vs systemic recurrence of tumours identified in this cohort?
Comparing the cohort over such a long time, is there any accounting for differences in surgical approach taken?
Why was neoadjuvant therapy given in some cases, I suspect these should be excluded?
Overall a robust study. Not unsurprising results, and more in-depth study of the tumour biology in these patients may be merited in future - this sis worthy of discussion
Author Response
Reviewer 3
Authors assess the relationship between clinicopathological factors and colorectal cancer recurrence instage 1 disease. This article is sound in its design and limited by the single centre nature of the work, as appreciated by the authors themselves. The findings of larger tumours, poor lymph node yield (a surrogate for inadequate surgery), and differentiation are not surprising.
Comments: Are there any predictors of local vs systemic recurrence of tumours identified in this cohort?
Response > Thank you for your detailed and kind review. We compared the each clinicopathologic factors included in Table 1 between local recurrence group (n=28), and systemic recurrence group (n=35). However, we did not find any predictive factors for a tumor recurrence in this cohort.
I agree with your opinion that the local and systemic recurrence is different, but the number of cases were not enough to analyze the risk factors. So we tried to analyze the total number of recurrences. But we also analyzed the detailed clinical findings after recurrence in Table 4. I think when we gather more case of local and systemic recurrence, we can analyze the detailed risk factors in further study.
Comparing the cohort over such a long time, is there any accounting for differences in surgical approach taken?
Response > The surgery for colon cancer was performed according to the principles including the cental node dissection (D2 or D3), no touch isolation, en bloc resection and surgical margin > 5-10cm. In the rectal cancer surgery, the procedures consisted of IMA high ligation, total mesorectal excision, distal margin > 5mm and intraoperative examination of frozen section for secure surgical margin. We additionally analyzed the association of operation period and the recurrence rate. The recurrence rate of stage I rectal cancer significantly decreased over the 15 year of this study period (3.7% in 2002-2006, 2.1% in 2007-2011, and 2.0% in 2012-2017). In the rectal cancer, we also newly found that the lymph node yield was significantly different by the operation period. The incidence of lymph node yield < 12 significantly decreased by the period as 56.6% in 2002-2007, 12.4% in 2008-2012, and 4.4% in 2013-2017 (p < 0.001).. When comparing the proportion of harvest LN <12 by the period of 2002-2007, 2008-2012, 2013-2017, the incidence of poor LN yield significantly decreased over the years. We suggest better lymph node yield has brought the decrease of recurrence rate in rectal cancer. This new findings following your comments were described in line 147-153, and 313-323 as follows:
line 147-153
“The recurrence rate difference by the operaton period (2002-2006, 2007-2011, 2012-2017) was analyzed. In the colon cancer group, using univariate analysis, the recurrence rate was 3.7% in 2002-2006, 2.1% in 2007-2011, and 2.0% in 2012-2017, respectively (p = 0.502). In the rectal cancer group, the recurrence rate was significantly different as 13.3%, 8.9%, and 5.0% with univariate analysis, respectively (p = 0.045). The statistical significance by the operation period in rectal cancer group did not present in the multivariate analysis.”
line 313-323
“This study presented the recurrence rate of stage I rectal cancer significantly decreased over the past 15 yeares of this study period. When comparing the lymph node yield, the incidence of lymph node yield < 12 significantly decreased by the period as 56.6% in 2002-2007, 12.4% in 2008-2012, and 4.4% in 2013-2017 (p < 0.001). The poor lymph node yield, or an inadequate lymph node examination due to lack of recognition for the importance of radicality based on lymph node retrieval may cause an underestimation of stage in early-stage CRC although the total mesorectal excision or high ligation of inferior mesenteric artery was performed. Howver, in the colon cancer, the recurrence rate did not differ by the operation period. This may result from the no difference in lymph node retrieval and standardization of surgical technique in colon cancer surgery.”
Why was neoadjuvant therapy given in some cases, I suspect these should be excluded?
Response > The patients who underwent neoadjuvant therapy were excluded at the beginning in this study. The discription about the exclusion criteria is in line 71-75.
Overall a robust study. Not unsurprising results, and more in-depth study of the tumour biology in these patients may be merited in future - this sis worthy of discussion
Response > The context you commented would make the discussion more enriched and suggest the in-depth study in the future. We added the description for the need of future in-depth study about tumor biology in line 303-305 as follows :
“The in-depth investigation for the tumor biologies of these patients who experience a tumor recurrence despite the early stage of cancer would be helpful to comprehend the disease and lead to the better clinical management.”
